# Artificial Intelligence-Assisted RFID Tag-Integrated Multi-Sensor for Quality Assessment and Sensing

**DOI:** 10.3390/s24061813

**Published:** 2024-03-12

**Authors:** Chenyang Song, Zhipeng Wu

**Affiliations:** Department of Electrical and Electronic Engineering, University of Manchester, Manchester M13 9PL, UK; songchenyang9454@hotmail.com

**Keywords:** RFID sensing, UHF RFID, RF energy harvesting, product quality assessment and sensing

## Abstract

Radio frequency identification (RFID) is well known as an identification, track, and trace approach and is considered to be the key physical layer technology for the industrial internet of things (IIoT). However, IIoT systems have to introduce additional complex sensor networks for pervasive monitoring, and there are still challenges related to item-level sensing and data recording. To overcome the shortage, this work proposes an artificial intelligence (AI)-assisted RFID-based multi-sensing technology. Both passive and semi-passive RFID tag-integrated multi-sensors are developed. The main contributions and the novelty of this investigation are as follows. A UHF RFID tag-integrated multi-sensor with a boosted charge pump is proposed; it enables high RF signal sensitivity and a long operational range. The whole hardware design, including the antenna and energy harvester, are studied. Moreover, a demonstration with real-world ham product sensing data is conducted. This work also proposes and successfully demonstrates the integration of machine learning algorithms, specifically the NARX neural network, with RFID sensing data for food product quality assessment and sensing (QAS). This application of machine learning to RFID-generated data for quality assessment is also a novel aspect of the research. The deployment of an autoregressive model with an exogenous input (NARX) neural network model, tailored for nonlinear processes, emerges as the most effective, achieving a root mean square error (RMSE) of 0.007 and an R-squared value of 0.99 for ham product QAS. By deploying the technology, low-cost, timely, and flexible product QAS can be achieved in manufacturing industries, which helps product quality improvement and the optimization of the manufacturing line and supply chain.

## 1. Introduction

Recently, internet of things (IoT) and radio frequency identification (RFID) technologies have set off a revolution in many industries [1,2,3]. These technologies offer potential capabilities that can improve the ecology of manufacturing industries by providing production quality assessment and sensing (QAS) and environmental conditions during the transportation, storage, and retailing process; these are ‘black box’ problems for conventional manufacturing industries [4]. Compared with the traditional product identification approach, such as barcode, RFID tag allows more information storage capacity to enable track and trace functions in the system [5]. Compared with the conventional QAS approaches, this work proposes an RFID-based QAS approach; the RFID-based systems provide multi-sensing capability without the sample process, laboratory conditions, or complex instruments. Also, RFID can enable passive sensing functions along with object identification functions. As a basic enabling technology of IoT, RFID also gives the system the chance to enable networked operations, which will enhance the system efficiency and accessibility. By deploying the RFID-based sensing technologies, the machine-to-machine (M2M) communication is enabled, which enables automatic and reasonable supply chain management (SCM), such as automatic control of environmental parameters, item sorting based on the production identification, and quality evaluation [6]. These benefits have led to a proliferation of the importance and investigation of IoT systems for manufacturing industry applications. Therefore, the improved RFID-based sensing technologies can be considered as an ultimate solution for modern smart manufacturing industries.

According to the operational principles, RFID electromagnetic sensing and RFID tag-integrated sensing are the two main approaches of RFID-based sensing. RFID electromagnetic sensing is based on tag antenna-based sensing (TABS), RFID backscattered signal sensing, or chipless RFID [7,8,9,10,11,12,13]. The TABS approach utilizes the change in RFID tag antenna characteristics when the RFID tag is attached to the material being tested. The RFID backscattered signal sensing concerns the change in backscattered signal features needed to evaluate the material being tested, such as the antenna resonance frequency, antenna gain, received signal strength indicator (RSSI), and received signal phase. Unlike normal RFID, chipless RFID uses electromagnetic structures to achieve data encoding [14]. The principle of chipless RFID sensing is the introduction of structures or materials that are sensitive to the sensing parameters. The encoded data can change according to the parameters. The design of the chipless RFID sensor heavily depends on the microwave materials. Moreover, the performance of RFID electromagnetic sensing heavily depends on the coupling condition between the RFID tag antenna, the environment, and the material being tested, which easily suffers from the change in environmental and external interference. Furthermore, a large number of studies have to be completed to establish the relationship between material features and antenna characteristics or backscattered signal features. Also, the power constraints of RFID electromagnetic sensors have a strong relationship with the environment and the sensing principle. In contrast, the RFID tag-integrated sensors have a relatively stable power constraint of about −30~−10 dBm RF sensitivity, which provides a long operation distance. On the other hand, the RFID tag-integrated sensor integrates the RFID tag chip, microcontroller unit (MCU), power management module, and sensors in one RFID tag, which enables sensing data interrogation in normal RFID tag read operations [15]. The RFID tag-integrated sensors can provide passive, flexible, stable, contactless, and low-cost multi-sensing capabilities, as well as data recording functions in various environments. Therefore, as an environment-monitoring and production QAS approach, the RFID tag-integrated sensors have the potential to be the critical technology of the next generation of the manufacturing line.

For RFID tag-integrated sensor design, both passive and semi-passive sensors have generated research interest. The semi-passive sensors are usually used in continuous and massive sensing applications, which require high power consumption. A stable power supply such as a cell battery is usually contained in the system. The passive sensors utilize the existing energy distributed in the environment to perform the sensing operations, such as solar energy harvesting, RF energy harvesting, and thermal energy harvesting [16,17,18,19,20]. For the RFID tag-integrated sensor in manufacturing industry applications, RF energy harvesting is the most reliable approach. Investigations involving the application of RFID tag-integrated sensors in wearable electronics, structure monitoring, indoor environment monitoring, and location and navigation have generated research interest [21,22,23]. However, there are still research blanks related to the design of a flexible and low-cost RFID tag-integrated sensor for the manufacturing line environment and product sensing and to the development of efficient product QAS methods. The analysis of load impacts in relation to RF energy harvesters is required for future passive RFID tag-integrated sensors. Moreover, accurate product QAS methods utilizing the RFID tag-integrated sensing data are necessary for the improvement of manufacturing lines. Although numerous studies have been conducted, the detailed discussions on hardware design, parametric studies, and the integration of AI with RFID sensing data have not been conducted. The main novelty and the contribution of this work are as follows: (1) UHF RFID tag-integrated multi-sensor design; (2) parametric study of RF energy harvester; (3) prototyping and design guide of passive and semi-passive RFID sensors; (4) AI-assisted QAS; (5) demonstration with ham product sensing data; and (6) integration of machine learning with RFID sensing data.

This work focuses on the RFID tag-integrated sensor for product QAS applications. First, the reference design of both passive and semi-passive RFID tag-integrated multi-sensors for product QAS is proposed. The sensor conforms to the EPCglobal Class 1 Generation 2 ultra-high frequency (UHF) RFID standard [24]. Then, the RF energy harvesting method is given. The study analyzes the impact of various loads and different harvester circuit parameters on the RF energy harvesting performance, which can support future investigation and designs. Moreover, this work proposes a T-matched meandered line antenna (MLA) and a folded dipole antenna for RFID tag interrogation and RF energy harvesting, respectively. A novel sensing data management structure which is compatible with the EPCglobal Class 1 Generation 2 standard is also proposed. Finally, the RFID tag-integrated passive and semi-passive sensors are implemented and validated. Artificial intelligence (AI) methods are used to perform product QAS. The sensors and methods are validated by measuring the food products, and the accurate results are validated.

This paper is organized as follows: Section 2 provides the design methodology for the passive and semi-passive RFID tag-integrated sensors for product QAS. The RF energy harvester, tag antenna, and rectenna of the passive RFID tag-integrated sensor are investigated in Section 3, and the parametric study is conducted. In Section 4, the RFID tag-integrated sensors are implemented and verified. Section 5 proposes AI-based product QAS methods using the sensor data. Finally, Section 6 concludes this work.

## 2. RFID Tag-Integrated Sensor Design

In order to sense the environmental information and product condition, two types of RFID tag-integrated sensors are designed: the passive RFID tag-integrated sensor and the semi-passive RFID tag-integrated sensor. To ensure dual-field (near-field and far-field) operation and the sufficient data capacity of the sensor, the EPCglobal Class 1 Generation 2 UHF RFID standard is selected for the sensor. In the UK, the RF operation follows the policy of an 865 MHz to 868 MHz operation frequency band and 2 W effective isotropic radiated power (EIRP).

The system framework of the passive RFID tag-integrated sensor is shown in Figure 1. The passive RFID tag-integrated sensor contains two main parts: the energy harvesting module and the RFID transceiver and integrated sensor module. The energy harvesting module collected the ambient RF energy and powered the RFID transceiver and integrated sensor module. To avoid data or RF collision, the design of this architecture has separate antennas for the RFID transceiver and RF energy harvester. Moreover, the dual-antenna design gets rid of the additional power management module, such as the RF combiner, divider, and switch, which reduces the system complexity and additional power cost.

During the operation, RF energy transmitted by the RFID reader is collected. After a series of RF to DC converters and voltage regulation, the DC power is carried out and fed to the other modules, such as the central controller, RFID tag chip, and sensors. The RF energy harvester requires sufficient energy-converting efficiency and has the capability to generate sufficient voltage and current for the integrated sensors to operate. In this design, the RF energy harvester consists of a rectifier antenna, matching circuit, RF to DC converter, charge pumper, and voltage regulator. The rectifier antenna receives the external RF energy. Due to the complex load situation, the input impedance of the RF energy harvester is not the ideal 50 Ohm. Therefore, an additional matching circuit is required to minimize the port reflection loss. The RF to DC converter can rectify the RF signal to the DC power supply. Due to the signal strength, the output voltage of the RF to DC converter may be insufficient for the sensing operations, which results in a low operation range and high RF signal requirements. Thus, a charge pumper is introduced to establish a charge–discharge cycle. In this way, the RF energy harvester can catch and store RF energy until sufficient power can be supplied to perform the sensing procedure, which makes the passive RFID tag-integrated sensor able to work even if the RF signal is weak. Finally, the voltage regulator generates the target voltage supply. The tag-integrated sensing module forms a sensing and data collection medium. Three main functional modules are required in the integrated sensing part, including the RFID transceiver module, sensor module, and central controller module. The RFID transceiver module consists of an RFID tag chip and a tag antenna. The sensors include both analogue and digital sensors to maintain multiple sensing capabilities. The low-power MCU is used to perform the sensing and data transfer process. As the item information storage medium, the RFID tag chip requires sufficient data storage capacity, especially for user data storage. The sensing data are recorded for QAS, and the QAS results are backed to the tags as the operation reference of the products. In order to achieve sensor data communication, a programmable RFID tag chip with an inter-integrated communication (I2C) interface is introduced. The design also has good extensibility.

Figure 2 shows the block diagram of the semi-passive RFID tag-integrated sensor. In contrast to the passive RFID tag-integrated sensor, the semi-passive sensor contains no RF energy harvester. An external power supply or battery is usually used. The semi-passive sensor can drive sensors with higher power consumption and can operate continuously, which is suitable for the environmental monitoring of the manufacturing line.

## 3. RF Energy Harvester and Sensor Module Design

### 3.1. RF to DC Converter

The easiest approach to converting an RF signal to a DC signal is to use a diode that filters the reverse current. Based on this idea, Cockcroft–Walton and Dickson J. F. Dickson proposed the well-known Cockcroft–Walton and Dickson multipliers [25,26]. The original aim of the voltage multiplier is to create low to high DC voltage conversion as a step-up DC-DC voltage booster. This work uses the voltage multiplier to achieve RF to DC conversion and conducts a parametric study for RF power harvesting in the UHF RFID. 

#### 3.1.1. Dickson Multiplier

According to the existing results, the Cockcroft–Walton voltage multiplier and the Dickson multiplier share similar RF energy harvesting performances. In order to keep a simple and easy-to-fabricate circuit architecture, the Dickson multiplier is applied. Figure 3 shows a four-stage Dickson multiplier. In the Dickson multiplier, two capacitors and two series diodes constitute one converting stage. The capacitors are connected between one diode and the DC output or the RF in. The RF incident wave has a fluctuation, and the input voltage thus varies between negative and positive. When the RF incident wave voltage is negative, the charge is accumulated at the cathode of the diodes D1, D2, D3, and D4. When the RF incident wave voltage changes to positive, the diodes D1′, D2′, D3′, and D4′ are activated and the charge is then gathered through these diodes. The charge is continuously carried to the next stages through this cycle when receiving RF input signals. The final output voltage is proportional to the amount of charge at the output node.

In the RF to DC conversion application, the conversion efficiency is heavily depending on the diode characteristics. Due to the high current direction switching speed in the UHF band, the diode is required to have a high switching speed. The Schottky diode, which is named after its inventor, Dr. Schottky, can fulfil this requirement. The forward conduction threshold voltage and forward voltage drop of the Schottky diode are both lower than the PN junction diode due to the lower barrier height. The reverse recovery time of the Schottky diode can be as low as 100 ps, which is much less than that of the *p*-n-type diode, which has the lowest recovery time of about 100 ns [27]. Hence, the voltage multiplier constructed with Schottky diodes is proposed as the key component with which to build the RF to DC converter.

#### 3.1.2. Parametric Study

The Dickson multiplier for the RFID tag-integrated sensor is then designed by following the operation principle. A few key factors, including the stage capacitance, RF incident power, the number of converting stages, and the load impedance, are considered when constructing the Dickson multiplier. The stage capacitors are the first concern. The only requirement of the capacitors is that they should be large enough for the charging–discharging cycles. In this application, 1pF capacitors are implemented. As the critical component of the Dickson multiplier, the choice of the Schottky diode is also very crucial. The mainstream Schottky diode products are investigated and compared. The chosen diodes should have the lowest possible turn-on voltage and the fastest switching time due to the high-frequency operation environment. Hence, HSMS-282x is selected. The incident power of the RF to DC converter is another important factor that affects the efficiency and output power. In the UK UHF RFID regulation, the maximum ERIP is 2 W. The incident power of the RF to DC converter is about −20 dBm to 20 dBm. The designed converter should have high power conversion efficiency in the power range. Finally, the load impedance also affects the output voltage and power. The RF to DC converter is required in order to have sufficient power and voltage to drive the overall circuits. To optimize the design, the parameters are investigated using the advanced design system (ADS) simulation.

First, the simulation is conducted to explore the relationship between the number of stages and the output voltage, power, and power conversion efficiency in the different load conditions. The simulation results are shown in Figure 4, Figure 5 and Figure 6. The power conversion efficiency is calculated as
(1)$$Efficiency=\frac{P_{o}}{P_{in}}\times 100\%$$
where *P_o_* and *P_in_* are the harvester output power and incident power, respectively. According to the results, the multiplier with less stages performs better in low incident power conditions.

Moreover, the simulation is conducted to evaluate the power conversion performance with respect to different loads. According to the results, the load resistance has limited impact on the power conversion performance when the load resistance is sufficiently large. The on-resistance of the MCU and sensors is large. Therefore, the impact of load can be neglected.

### 3.2. SOI-Based Charge Pumper

According to Figure 4a and Figure 7a, the output voltage is quite low when the incident power is low. When the voltage is lower than the lowest operation voltage of the voltage regulator, the harvester cannot work. Also, the continuous DC output power may not be large enough to support a whole sensor process. Therefore, the harvester has to be set closely to the RFID reader antenna to obtain sufficient output voltage. To avoid this drawback and to increase the power conversion efficiency, a boost charge pump is introduced (Figure 8).

The charge pump deployed fully depleted silicon-on-insulator (SOI) technology, which can enable ultra-low-power operation and increase energy efficiency. Utilizing SOI technology in transistor fabrication, where a silicon layer is isolated from the substrate by an insulating layer, significantly boosts the efficiency of a charge pump [28]. The lowest input voltage limit of charge pump is 0.35 V. During the RF energy harvesting, the oscillation circuit in the charge pump operates when the DC voltage output by the Dickson multiplier is larger than 0.35 V. Hence, a high-sensitivity RF energy harvester with a long operation range is obtained. The electric power for the DC voltage step-up is then charged into a storage capacitor which has a relatively large capacitance. The sensor system operates at a 1.8 V level. The storage capacitor starts to discharge when its voltage reaches 2.4 V. The discharge stops when the voltage has decreased to 1.8 V, and the capacitor is charged again. During the discharge process, a low-power voltage regulator regulates the output voltage to 1.8 V. In this way, a charge–discharge cycle is realized. The discharge time depends on the power consumption and the capacitance of the storage capacitor.

### 3.3. T-Matched Folded Dipole Antenna

The power-admitting efficiency of the antenna, *η*, decides the total available power of the RF to DC converter. The gain of the antenna heavily depends on its directivity, and the relationship between them under the same electric field can be presented by
(2)$$G(\theta ,\varphi )=\frac{P_{\mathrm{in}0}}{P_{\mathrm{in}}}=\frac{P_{r}}{P_{\mathrm{in}}}\cdot \frac{P_{r0}}{P_{r}}=\eta \cdot D(\theta ,\varphi )$$
where *P_r_*_0_ and *P*_in0_ are the source antenna radiated power and input power, respectively; *θ* and *ϕ* are the azimuth and elevation degrees of the spherical polar coordinate; and *D*(*θ*, *ϕ*), *G*(*θ*, *ϕ*) are the antenna directivity and gain, respectively [29]. Therefore, a trade-off between the directivity and gain of the antenna needs to be considered to ensure a higher energy reception efficiency over a wide angle.

In light of these considerations, this work develops a T-matched folded dipole antenna for RF energy harvesting [30]. The antenna is designed on a 1.6 mm thick FR-4 substrate (εr = 4.4, tanδ = 0.020) and has a compact size of 75 mm × 60 mm. The antenna has a gain of about 5 dB with about an HPBW of 80 degrees, which eases the deployment of the sensor. In addition, the folded arm design gives the antenna a circuit deployment area, as shown in Figure 9, which results in a compact passive RFID tag-integrated sensor design.

### 3.4. T-Matched MLA for RFID Tag

The tag antenna is the other part of the RFID module which enables wireless communication between the RFID chip and reader. To realize RFID operation in both the near field and far field, a dipole-like antenna is commonly used for RFID tag antenna. For the RFID tag, the two critical design difficulties are antenna size and impedance due to the size limit of the RFID tag and the complex impedance of the RFID tag chip. To reduce the size of the UHF RFID antenna, there are two effective approaches; these are the meandering and the inverted-F configurations. They are usually deployed in dipole and monopole antennas. However, the inverted-F architecture antenna has a stereoscopic structure. For the RFID tag-integrated sensor, it gives unneglectable thickness to the tag and increases the fabrication complexity. Therefore, the MLA is used as the tag antenna. Meandering refers to the folding of the dipole antenna arms and the reduction in their size. The parallel transmission lines and turns form equivalent capacitance and inductance and change the impedance of the antenna. Compared to typical dipole antennas with straight arms, MLAs have a more compact size. Moreover, the capacitance and inductance introduced by meandering benefit impedance matching. However, the bandwidth and directivity of MLAs are also affected.

This work uses the UCODE I2C tag fabricated by the NXP Semiconductor with programmable features. The UCODE I2C tag has a characteristic impedance of 13.8 − j210 Ω. Therefore, the tag antenna should have an impedance of 13.8 + j210 Ω, which is a conjugate matched to the tag chip. A T-matched half-wave dipole MLA is designed as the tag antenna. Figure 10 shows the designed MLA, and Table 1 illustrates the detailed dimensions of the MLA. The antenna is designed on an FR-4 substrate with a relative dielectric constant ε_r_, substrate thickness H_s_, and copper thickness H_c_. The overall length of the conductor is near half-wavelength at 866.5 MHz, which makes the antenna a half-wave dipole antenna.

The designed MLA is simulated using CST Microwave Studio, and the prototyped antenna is measured using an E5071 programmable vector network analyzer in an anechoic chamber. Figure 11 shows the simulated and measured S11 of the fabricated MLA. The fabricated MLA has an impedance bandwidth (S11 < −10 dB) of 9.20 MHz (1.04%, 865.01–874.21 MHz), which can cover the UK UHF RFID band. Figure 12 shows the simulated far-field radiation pattern of the MLA on the E-plane and H-plane. The red line indicates the far-field gain. The blue line shows the half power beam width (HPBW). The MLA has a maximum gain of 0.677 dB, with a 0.2 dB maximum gain variation on the H-plane. The HPBW is 87 degrees, which can fulfil the RFID tag-integrated sensor applications.

### 3.5. MCU and Sensor Units

In addition to the mentioned designs, off-the-shelf low-power MCUs and sensors are applied to perform the sensing process. The key components are shown in Table 2.

## 4. Implementation and Verification

To evaluate the performance of the designed RFID tag-integrated sensors, both the passive and semi-passive sensors are fabricated. The prototypes are shown in Figure 13.

To evaluate the RF energy harvester performance, the passive RFID tag-integrated sensor is set up, as shown in Figure 14a. The RF signal is transmitted by the Alien 9900+ UHF RFID reader with a 30 dBm output and a 5 dBi circularly polarized UHF RFID reader antenna. The passive RFID tag-integrated sensor is moved away from the reader antenna. An MCU with a built-in ADC is used to measure the voltage of the storage capacitor. The time stamp is added to the sampled voltage. Then, a time–voltage matrix is recorded. The single-cycle sensing time, the first-time charging time (voltage of storage capacitor rising from 0 V to 2.4 V), and the re-charging time (voltage of storage capacitor rising from 1.8 V to 2.4 V) are recorded. According to the results, one sensing cycle takes 195 ms, and the discharge time is sufficient to support the sensing cycle. At the distance of 200 cm, the charging time of the cold start is 70.26 s, while the re-charging time is 12.01 s. The maximum operation range is about 350 cm. When measuring the power conversion efficiency of the RF energy harvester, the output power of the RFID reader is gradually decreased from 20 dBm to −16 dBm. The calculated efficiency is shown in Figure 14b. The maximum power conversion efficiency is about 31%. For the semi-passive design, the sensor power is guaranteed by a battery or an external power supply. Therefore, the operation range of the semi-passive design is same as that of the RFID tag detection range, which is 8 m in this case.

To verify the sensing capability of the prototyped sensor and to collect data for the product QAS, the semi-passive RFID tag-integrated sensor is set to measure ham products, as shown in Figure 15a. The temperature, relative humidity, ambient light, equivalent CO_2_ concentration, equivalent total violate organic compound (eTVOC) concentration, and color are obtained. The results are illustrated in Figure 15b–i. As shown in Figure 15f, there is a significant increase in eTVOC at the 112th hour of ham storage, which represents the massive reproduction of microbes; at this point, the product is fully spoiled. This time is before the indicated shelf life. Therefore, RFID tag-integrated sensing is meaningful. Accordingly, the proposed RFID tag-integrated sensor can provide product-level and environment-level sensing for manufacturing industries.

## 5. AI-Based QAS

Due to the features of RFID, the sensing data can be recorded in the user bank of the RFID tag memory. This offers opportunities to obtain the product condition and the environmental condition histories along the manufacturing line. With the support of sufficient data, a model of product condition histories and product quality can be established using AI methods. In this way, timely and accurate nondestructive product QAS can be conducted.

Based on the ham sensing experiments, this work proposes an AI-based food product QAS approach as a demonstration. To prepare the training dataset, multiple experiments are repeated. The ham products are stored in the different conditions until they are spoiled, such as at room temperature and in the refrigerator. The sensing is performed every 5 min by scanning the RFID tag-integrated sensor. With the data collected during the ham tests, the model for ham QAS is built with the AI-based sensor fusion algorithms. The measured data form a time series dataset that contains multiple input sensory variables, including the temperature in degrees, RH as a percentage, TVOC concentration in ppb, CO_2_ concentration in ppm, RGB information, ambient light intensity in lux, color temperature in Kelvin, and storage time in days. The output dependent variable is the shelf life. The multi-sensing time series data can reflect the quality and the quality change trend. Therefore, the multi-sensing data can be utilized to assess the quality of food products using machine learning approaches. The QAS process is to build regression models between these data and the shelf life. To establish the training dataset, 60% of the measured product data is used as a training dataset. The validation dataset and test dataset each take 20% of the measured product data. Five-fold cross-validation is used during validation. The regression algorithms, including linear regression (LR), regression tree, support vector regression (SVR), the feedforward neural network (FFNN), and the nonlinear autoregressive model with the exogenous input (NARX) neural network are investigated. The block diagram of the NARX neural network is shown in Figure 16 [40]. NARX is a type of recurrent neural network. As shown in Figure 16, compared with conventional RNN, the autoregressive part (*y*) of the NARX neural network considers past values of the time series itself in addition to the time series exogenous input (*x*). *d* represents the delay of the time series data. A three-layer NARX neural network is used. Also, the NARX model incorporates a nonlinear mapping function that captures complex relationships within the data. This nonlinear mapping allows the model to handle systems with nonlinearity, making it more flexible in representing diverse dynamics. As the QAS process is a nonlinear process with RFID tag recorded time series data, the NARX model is suitable. The trained models are used to perform the QAS of unseen ham product data. The corresponding storage time-evaluated shelf life curves are obtained, and the residuals are calculated; these are shown in Figure 17a and Figure 17b, respectively.

According to the results, the models can evaluate ham product qualities to a certain extent, and the NARX neural network has the best performance due to the utilization of time series data and recurrent input. To quantitatively analyze the QAS accuracy, the RMSE and R-square of the NARX neural network QAS model is calculated. The equations can be written as
(3)$$RMSE=\sqrt{\sum_{i=1}^{n}\frac{{\left(\hat{y_{i}}-y_{i}\right)}^{2}}{n}}$$
(4)$$R^{2}=1-\frac{\sum_{i=1}^{n}{\left(y_{i}-\hat{y_{i}}\right)}^{2}}{\sum_{i=1}^{n}{\left(y_{i}-\bar{y}\right)}^{2}}$$
where $\hat{y_{i}}$ is the evaluated value, *y_i_* is the actual value, and *n* is the number of samples. *R*^2^ is the coefficient of determination that is in proportion to the variation in the predicted output variable from the input variables, which is proposed for evaluating the regression. The value of R2 is between 0 and 1 [41]. The closer the *R*^2^ is to 1, the more reasonable the model’s interpretation of the regression data is. The MLR, SVR (radial basis function kernel), regression tree, and FFNN have RMSEs of 1.0945, 0.6662, 0.5566, and 0.5405, respectively, which are unacceptable for food products. For the NARX neural network model, the RMSE and R-squared are 0.007 and 0.99, respectively, which shows that the RFID tag-integrated sensors can strongly and efficiently support product QAS in the manufacturing industries.

Table 3 shows the comparison of the recent investigations of the RFID-enabled sensors. According to the studies, RFID tag-integrated sensors have longer distances because the RFID tag-integrated sensor has high RF signal sensitivity while the other approaches are easily affected by the environment. In addition, the RF energy is more reliable than solar or piezoelectric power conversion. Compared with the existing studies, this study proposes an RFID tag-integrated multi-sensor and discusses full-process hardware design. Both passive and semi-passive designs are given. Through comparisons with the existing investigations, the AI-based food product QAS method based on RFID sensing data is demonstrated.

## 6. Conclusions

In conclusion, this investigation contributes a robust design methodology for UHF RFID tag-integrated multi-sensors, underpinned by advancements in RF energy harvesting and sensor integration. The prototypes of both the passive and the semi-passive RFID tag-integrated sensors showcase their potential for passive, flexible, and cost-effective multi-sensing solutions, offering stability and contactless operation. Importantly, the introduction of an SOI-based boosted charge pump in the RF energy harvester significantly enhances operational range and efficiency, making these sensors viable for diverse manufacturing applications.

Additionally, this work introduces an innovative AI-assisted quality assurance system (QAS) based on RFID tag-integrated sensors. The approach involves the creation of machine learning regression models, trained with a comprehensive dataset from experiments, and has a particular focus on ham products. Among the models investigated, the nonlinear autoregressive model with an exogenous input (NARX) neural network emerges as exceptionally effective, achieving remarkable accuracy with low root mean square error (RMSE) and high R-squared values. This underscores the potential of RFID tag-integrated sensors in providing timely and precise product quality assessment for manufacturing industries. The proposed methodology not only facilitates real-time monitoring of product conditions but also allows historical tracking through RFID tag memory. This establishes a foundation for predictive modeling and quality control and addresses the evolving needs of modern manufacturing. The comparative analysis with recent RFID-enabled sensors underscores the superior performance of RFID tag-integrated sensors, particularly in terms of distance and reliability.

In summary, the amalgamation of cutting-edge RFID sensor technology and artificial intelligence presented in this work signifies a significant advancement in manufacturing quality control. The RFID tag-integrated multi-sensor system, coupled with AI-driven QAS, stands poised to revolutionize the industry by providing an accurate, nondestructive, and timely approach to ensuring product quality throughout the manufacturing process.

## Figures and Tables

**Figure 1 sensors-24-01813-f001:**
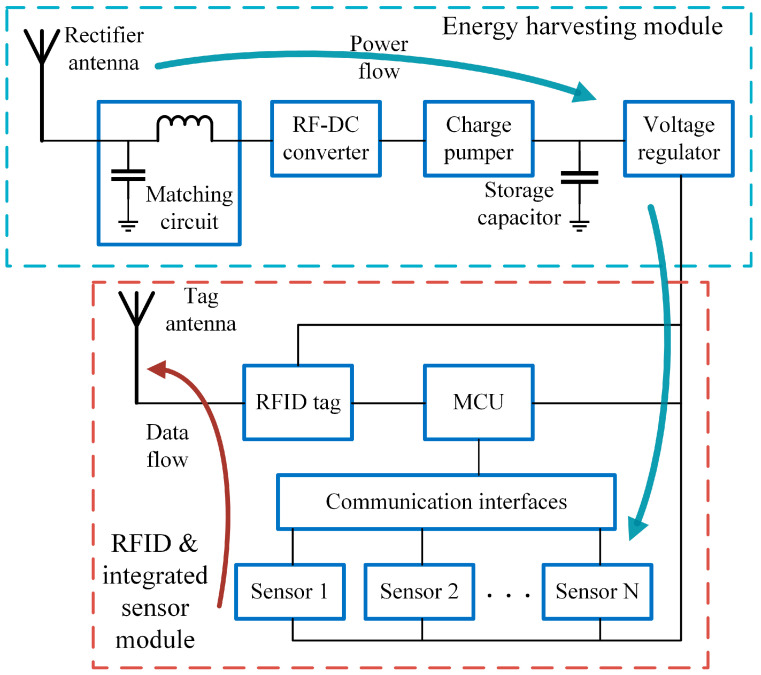
System framework of the passive RFID tag-integrated sensor.

**Figure 2 sensors-24-01813-f002:**
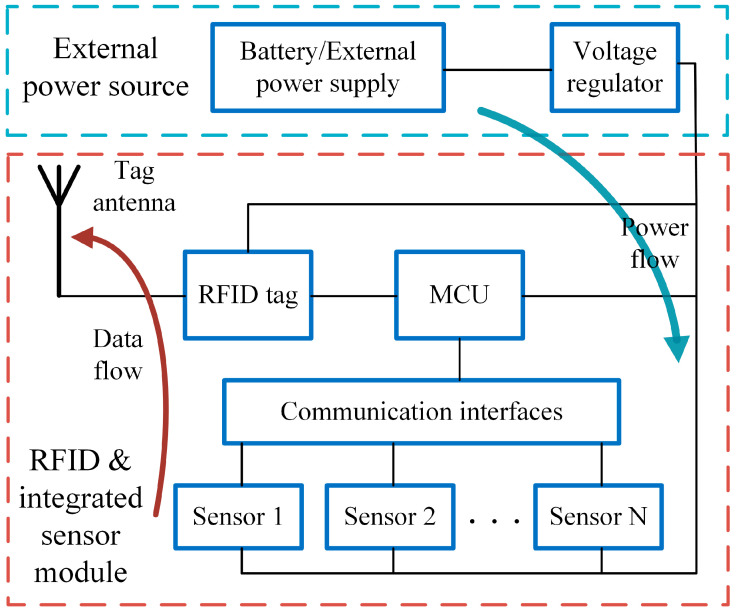
System framework of the semi-passive RFID tag-integrated sensor.

**Figure 3 sensors-24-01813-f003:**
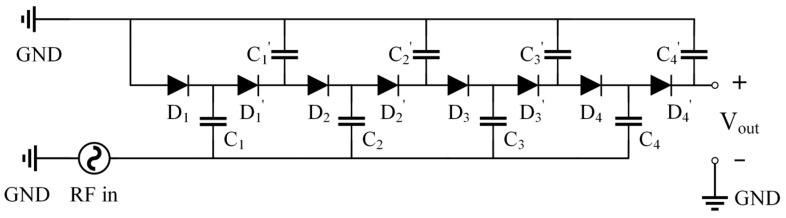
Four-stage Dickson multiplier.

**Figure 4 sensors-24-01813-f004:**
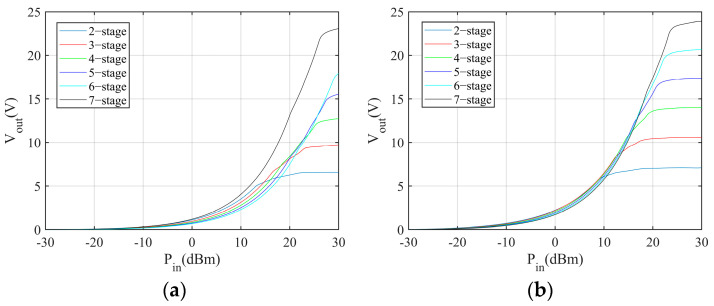
Output voltage of Dickson multipliers with different numbers of stages: (**a**) 5 kΩ load, (**b**) 15 kΩ load.

**Figure 5 sensors-24-01813-f005:**
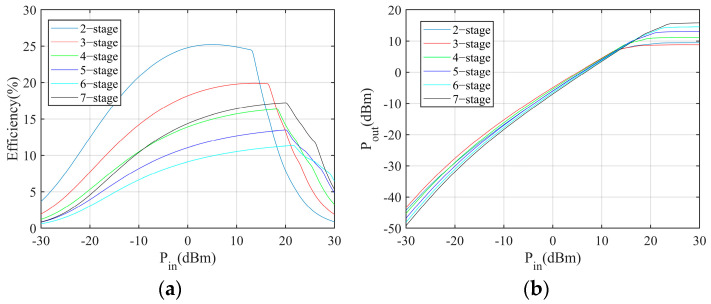
Output power of Dickson multipliers with different numbers of stages: (**a**) 5 kΩ load, (**b**) 15 kΩ load.

**Figure 6 sensors-24-01813-f006:**
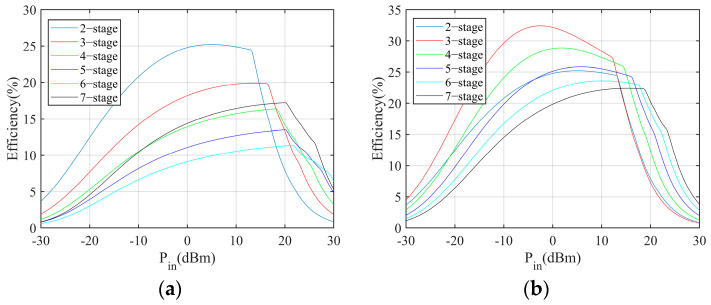
Power conversion efficiency of Dickson multipliers with different numbers of stages: (**a**) 5 kΩ load, (**b**) 15 kΩ load.

**Figure 7 sensors-24-01813-f007:**
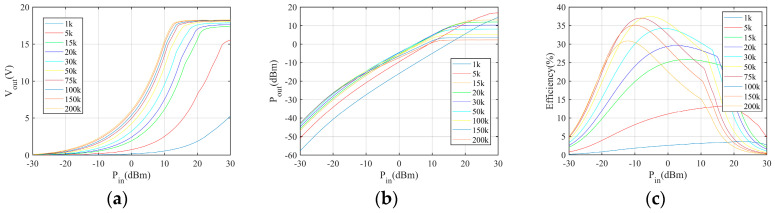
Performances of the 5−stage Dickson multiplier: (**a**) output DC voltage, (**b**) output DC power, (**c**) power conversion efficiency.

**Figure 8 sensors-24-01813-f008:**
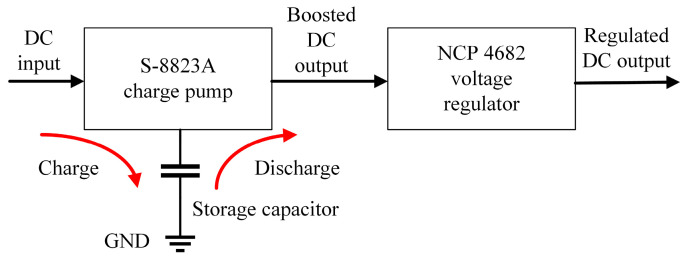
The DC power management module including charge pump and voltage regulator.

**Figure 9 sensors-24-01813-f009:**
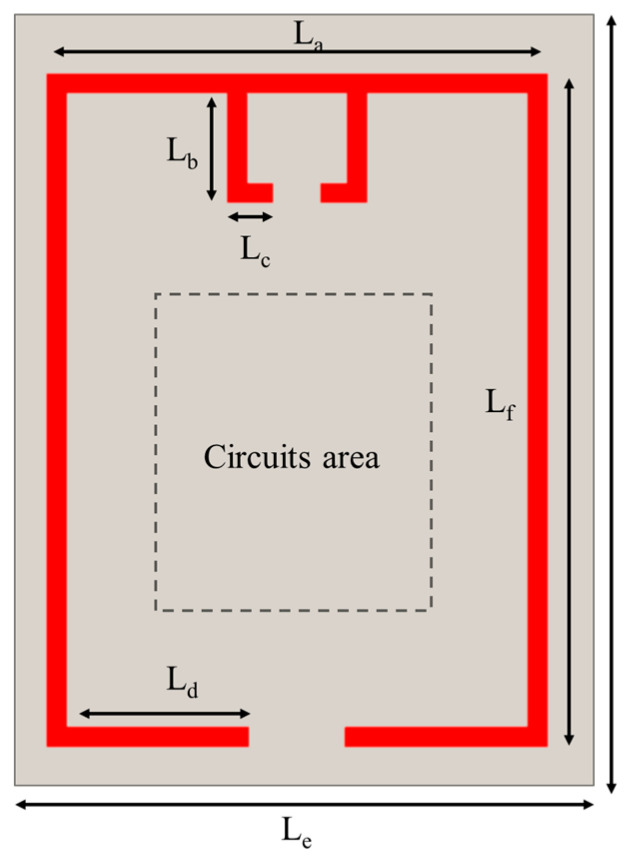
The layout of RF energy harvesting antenna.

**Figure 10 sensors-24-01813-f010:**
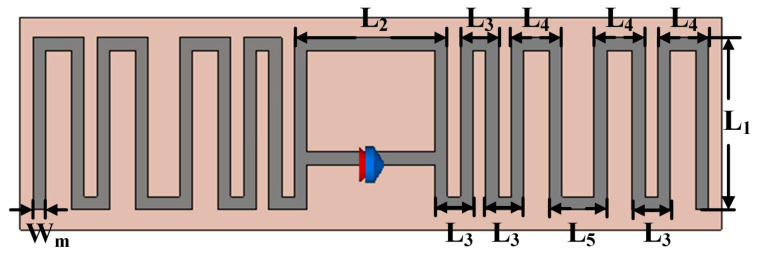
T-matched MLA for RFID tag-integrated sensor application.

**Figure 11 sensors-24-01813-f011:**
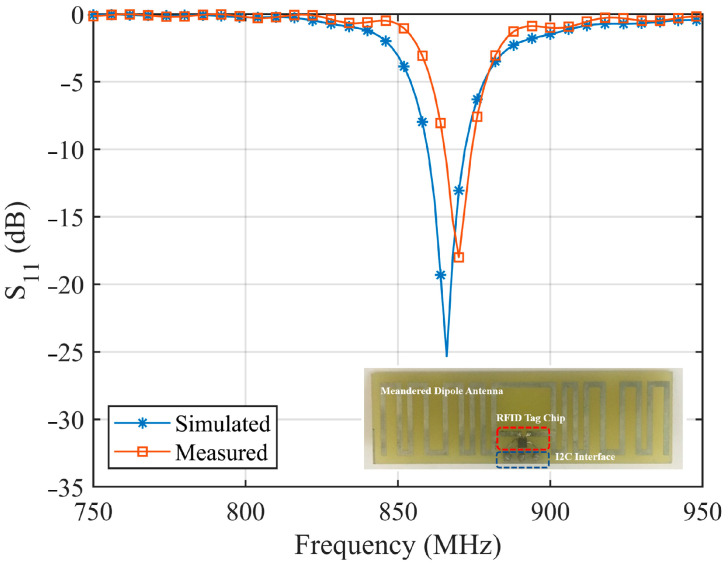
Simulated and measured S_11_ of the designed MLA.

**Figure 12 sensors-24-01813-f012:**
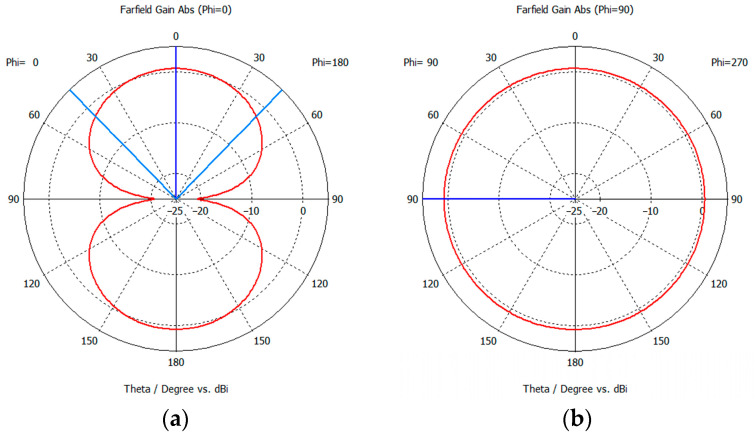
Radiation pattern of the designed MLA: (**a**) E-plane, (**b**) H-plane.

**Figure 13 sensors-24-01813-f013:**
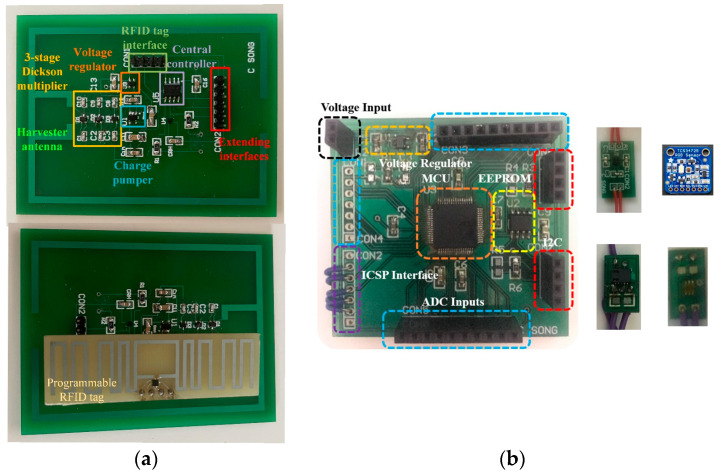
Fabricated RFID tag-integrated sensors: (**a**) passive design, (**b**) semi-passive design.

**Figure 14 sensors-24-01813-f014:**
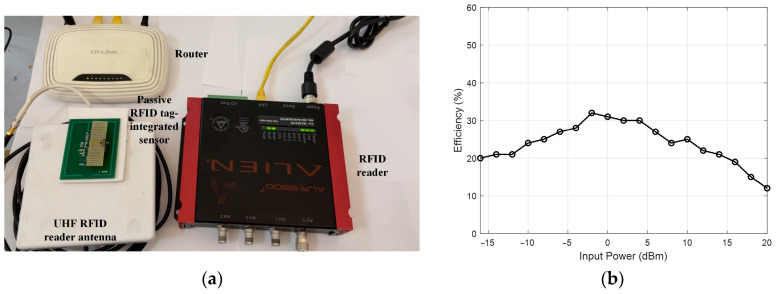
RF harvester performance evaluation: (**a**) experimental setup, (**b**) energy harvesting efficiency.

**Figure 15 sensors-24-01813-f015:**
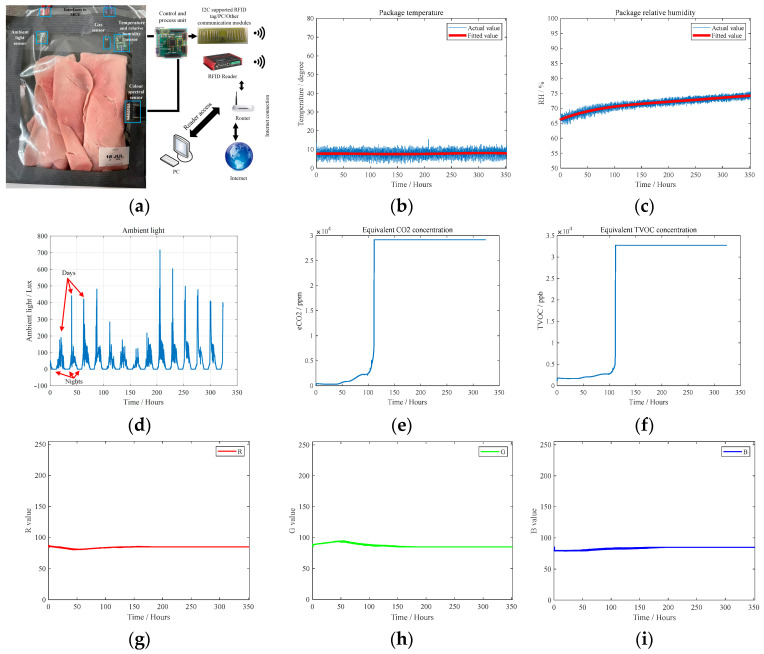
The RFID tag-integrated sensing results when measuring ham products: (**a**) experiment setup, (**b**) temperature, (**c**) relative humidity, (**d**) ambient light density, (**e**) CO_2_ concentration, (**f**) TVOC concentration, (**g**–**i**) color information (R,G,B).

**Figure 16 sensors-24-01813-f016:**
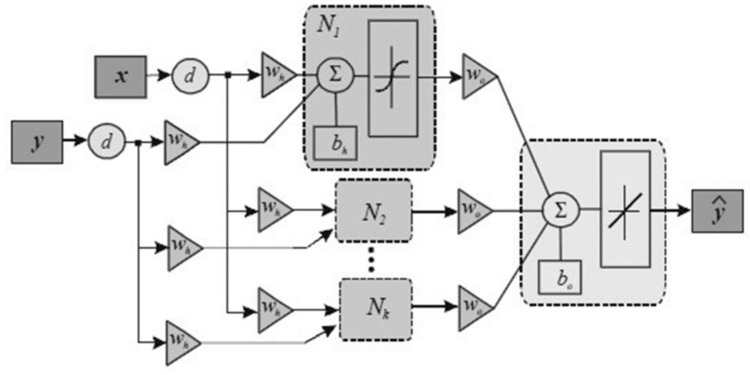
Block diagram of nonlinear autoregressive model with exogenous input (NARX) neural network.

**Figure 17 sensors-24-01813-f017:**
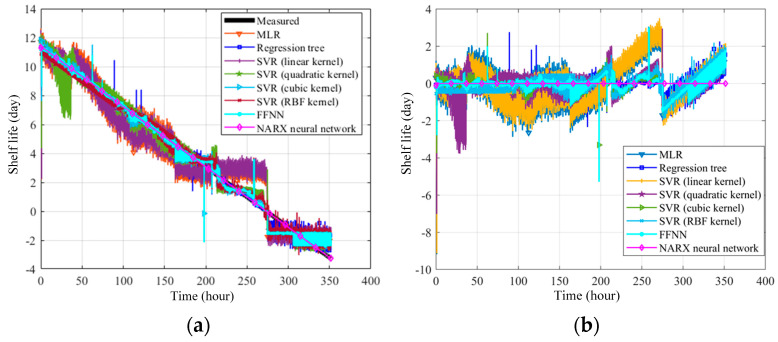
Ham product QAS results: (**a**) evaluated shelf life with respect to the storage time, (**b**) residual between evaluated shelf life and actual shelf life.

**Table 1 sensors-24-01813-t001:** Design parameters of the T-matched MLA.

**Parameters**	**L_1_**	**L_2_**	**L_3_**
Values	18 mm	12 mm	3 mm
**Parameters**	**L_4_**	**L_5_**	**W_m_**
Values	4 mm	4.5 mm	1 mm
**Parameters**	**ε_r_**	**H_s_**	**H_c_**
Values	4.4	1.6 mm	0.035 mm

**Table 2 sensors-24-01813-t002:** Key components list.

Components	Models
Schottky diode ^#^	HSMS2822 (Avago Technologies, Palo Alto, CA, USA) [31]
Stage capacitor ^#^	330µF, Multi-layer Ceramic Capacitor
Storage capacitor ^#^	1 pF, Multi-layer Ceramic Capacitor
Boost charge pump ^#^	S8823A (ABLIC, Tokyo, Japan) [32]
Voltage regulator	NCP4682 (ON Semiconductor, Scottsdale, Arizona, USA) [33]
MCU	PIC12LF1840 (Microchip Technology, Chandler, Arizona, USA) [34]
Temperature sensor	HDC2010 (Texas Instruments, Dallas, Texas, USA) [35]
Humidity sensor	HDC2010 (Texas Instruments)
Ambient light sensor	TSL25721 (AMS OSRAM, Premstaetten, Austria) [36]
RFID tag IC	NHS3100UCODEADK (NXP Semiconductor, Eindhoven, The Netherlands) [37]
* Gas sensor	CCS811 ((AMS OSRAM) [38]
* Color sensor	TCS34725 (AMS OSRAM) [39]

* The component is used in semi-passive design only; ^#^ The component is used in passive design only.

**Table 3 sensors-24-01813-t003:** Comparison of recent studies on RFID-enabled sensors.

Work	Sensing Parameter	Power Source/ Technology	Peak Power Conversion Efficiency	Operation Range	Post-Processing
[28]	Temperature	RF, passive /Tag-integrated sensor	16%	5 m	No
[42]	Soil moisture, temperature	RF, passive /Tag-integrated sensor	n.a.	2 cm	Yes
[23]	Localization	RF, passive /Backscatter signal	n.a.	n.a.	Yes
[43]	n.a.	RF, passive /Tag-integrated sensor	60%	n.a.	No
[44]	n.a.	Piezoelectric, active /Tag-integrated sensor	n.a.	n.a.	No
[45]	Accelerometer, temperature	RF, passive /Tag-integrated sensor	n.a.	5.5 m	No
[46]	Vibration Frequency	RF, passive /TABS	n.a.	1.5 m	No
[17]	Temperature	Solar, passive /Tag-integrated sensor	n.a.	1.5 m	No
This work	Temperature, humidity, ambient light, gas, color	Battery, semi-passive /Tag-integrated sensor	n.a.	8 m	Yes
This work	Temperature, humidity, ambient light	RF, passive /Tag-integrated sensor	31%	3.5 m	Yes

Comparison of power conversion efficiency and operation range is not strictly fair because of the differences in load, operation frequency, and transmitting power, among other considerations.

## Data Availability

The raw data supporting the conclusions of this article will be made available by the authors on request.

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
