# Peer review of "Artificial Intelligence-Assisted RFID Tag-Integrated Multi-Sensor for Quality Assessment and Sensing"

_sensors, 2024, doi:10.3390/s24061813_

Round 1

Reviewer 1 Report

Comments and Suggestions for Authors

sensors-2870728

Title: Artificial Intelligence Assisted RFID Tag-integrated Multi-sensor for Quality Assessment and Sensing

Indeed, the manuscript is well-written and easy to follow. Some points need to be known.

-What do the authors mean by “SoI-based boot charge pump” in line 414?

- How does the authors measure timings “70.26 s” from cold start and “12.01 s” for further sensing cycles?

The blog diagram shown in Figure 16 needs to be explained more clearly.

-It will be good to include a complete picture of the experimental setup instead of the block diagram as depicted in Figure 16.

-The naming of parts in Figure 15 a is not clear.

-Please name the components shown in Figure 14.

-Table2: Please mention the type of capacitors used in this work (330uF, 1pF).

Adding the real picture of an experimental setup showing all the sensors and controllers mentioned in Table 2 will be good.

-The novelty of the work should be clearly highlighted (in the abstract and the conclusions).

-It is better to list a comparison table to compare results with previous work.

Comments on the Quality of English Language

Minor editing of the English language is required.

Author Response

  • Responses to Reviewer 1

Comments to the Author:

Indeed, the manuscript is well-written and easy to follow. Some points need to be known.

  • What do the authors mean by “SoI-based boot charge pump” in line 414?

Comment is well taken. As described in Section 3.2, SOI stands for Silicon-on-Insulator. The introduction of SOI-based charge pump can increase the energy efficiency and enable ultra-low power operation.

Utilizing Silicon-On-Insulator (SOI) technology in transistor fabrication, where a silicon layer is isolated from the substrate by an insulating layer, significantly boosts the efficiency of a charge pump for several reasons. Firstly, it diminishes capacitive coupling effects between transistors, reducing charge transfer losses. Additionally, SOI lowers the breakdown voltage of transistors, enabling charge pumps to operate efficiently at lower voltages. The technology also mitigates substrate effects, such as the substrate effect, by virtue of the insulating layer. This aids in minimizing charge scattering in the substrate, contributing to improved charge pump efficiency. Furthermore, the relatively thin silicon layer in SOI technology reduces impedance, enhancing the charge pump's overall efficiency. In essence, SOI technology enhances charge pump performance by addressing capacitive coupling, breakdown voltage, substrate effects, and impedance, making it particularly effective in low-power and low-voltage environments.

SOI chips consume 20% less power and can operate up to 15% quicker than bulk CMOS-based devices. Low noise and high-quality passives are other benefits of SOI involving RF and mixed signals applications. The used charge pump implements fully depleted SOI technology to enable ultra-low voltage operation. In the manuscript, the “SoI-based boot charge pump” should be “SOI-based boosted charge pump”, which has been corrected. Also, the SOI technology is introduced more clearly.  

  • How does the authors measure timings “70.26 s” from cold start and “12.01 s” for further sensing cycles?

Comment is well taken. An MCU with build-in ADC is used to measure the voltage of storage capacitor. The time stamp is added to the sampled voltage. Then a time-voltage matrix is recorded. When measuring cold start time, the storage capacitor, stage capacitors, and bypass capacitors are discharged. The time between the RF power transmission start and the storage capacitor reaches its discharge voltage (2.4 V) is recorded as the cold start time. The time between the storage capacitor discharge stops and the next discharge start (recharging from 1.8 V to 2.4 V) is recorded as the time for further sensing cycle. The measurement process is revised in more detail in the Section 4.

  • The blog diagram shown in Figure 16 needs to be explained more clearly.

Comment is well taken. The blog diagram in Figure 16 is explained more clearly in Section 5. The added content is also shown in this letter as below.

As shown in Figure 16, comparing with conventional RNN , the autoregressive part (y) of NARX neural network considers past values of time series itself in addition to time series exogenous input (x). d represents the delay of time series data. A three-layer NARX neural network is used. Also, the NARX model incorporates a nonlinear mapping function that captures complex relationships within the data. This nonlinear mapping allows the model to handle systems with nonlinearity, making it more flexible in representing diverse dynamics. As the QAS process is nonlinear process with RFID tag recorded time series data. The NARX model is suitable.

  • It will be good to include a complete picture of the experimental setup instead of the block diagram as depicted in Figure 16.

Figure 16 is the block diagram of the proposed QAS model. The experimental setup is shown in Figure 15 (a).

  • The naming of parts in Figure 15 a is not clear.

Comment is well taken. Figure 15 (a) is revised to show the naming of parts clearly.

  • Please name the components shown in Figure 14.

Comment is well taken. Name of the parts in Figure 14 are added.

  • Table2: Please mention the type of capacitors used in this work (330uF, 1pF).

Comment is well taken. They are both multi-layer ceramic capacitor. The type is now mentioned in Table 2.

  • Adding the real picture of an experimental setup showing all the sensors and controllers mentioned in Table 2 will be good.

Comment is well taken. Figure 14 shows the experimental setup for passive design while Figure 15 (a) shows the experimental setup for semi-passive design. Some sensors are only deployed in semi-passive design. Therefore, Figure 15 (a) shows all the sensors used in this demonstration.

  • The novelty of the work should be clearly highlighted (in the abstract and the conclusions).

The abstract and conclusion are revised to highlight the novelty and main contributions of this work. The revised parts can be found in the red text in abstract and conclusion.

  • It is better to list a comparison table to compare results with previous work.

A comparison is conducted in Table 3. This work is compared with recent RFID-enabled sensors. The performances are discussed.

  • Comments on the Quality of English Language:

Minor editing of the English language is required.

Comment is well taken. The English editing of the manuscript is polished.

Reviewer 2 Report

Comments and Suggestions for Authors

The authors presented their work, "Artificial Intelligence Assisted RFID Tag-integrated Multi-sensor for Quality Assessment and Sensing", which seems to be significant and important from a research point of view. However, some minor corrections are required to be done from the author's side, especially with the addition of performance comparisons with the existing works in the same field of interest. The rest of it looks good.

Author Response

  • Responses to Reviewer 2

Comments to the Author

The authors presented their work, "Artificial Intelligence Assisted RFID Tag-integrated Multi-sensor for Quality Assessment and Sensing", which seems to be significant and important from a research point of view. However, some minor corrections are required to be done from the author's side, especially with the addition of performance comparisons with the existing works in the same field of interest. The rest of it looks good.

Comments are well taken. Performance with the existing works in the same field of interest are compared in Table 3.

Reviewer 3 Report

Comments and Suggestions for Authors

The authors have presented a novel method to assess product shelf life with tag-integrated sensor system using AI-method. The overall presentation is acceptable, but improvement is required before considering publication.

1. The tag-integrated sensors are advantageous compared with TABS in the review, how about the energy constraint for RFID systems? And a review for chipless sensing systems is missing for comparing different RFID sensing method. By the way, ref.19 is chipless RFID which is not correctly referenced. 

2. The AI-based shelf life estimation algorithm is not well presented. How are the data split as training group and validation group? What is the performance advantage for NARX method compared with others in quantitative results? 

Author Response

  • Responses to Reviewer 3

Comments to the Author

The authors have presented a novel method to assess product shelf life with tag-integrated sensor system using AI-method. The overall presentation is acceptable, but improvement is required before considering publication.

  • The tag-integrated sensors are advantageous compared with TABS in the review, how about the energy constraint for RFID systems? And a review for chipless sensing systems is missing for comparing different RFID sensing method. By the way, ref.19 is chipless RFID which is not correctly referenced.

Comments are well taken. First, the power constraint of RFID tag-integrated sensor is discussed in Section 1. In addition, the chipless RFID sensing is also discussed in Section 1. RFID tag-integrated sensing has a more stable power constraint while RFID electromagnetic sensing is heavily depending on the environment and sensing principle. The ref.19 (now ref 22) has been corrected.

  • The AI-based shelf life estimation algorithm is not well presented. How are the data split as training group and validation group? What is the performance advantage for NARX method compared with others in quantitative results?

Comments are well taken. For the measured product data, 60% data is used for model training, 20% is used for cross validation, and 20% data is reserved for model test.

The Section 5 is revised. First, the advantage for NARX method is briefly analyzed in theory. The performance advantage of NARX in quantitative results is then discussed by comparing to the results of other mentioned models. The following table shows the performance comparison.

Table 1 Performance of different models

Models

RMSE

R2

MLR

1.0945

0.9302

Regression tree

0.5566

0.9821

SVR_Linear

1.1315

0.9300

SVR_Quadratic

0.7038

0.9741

SVR_Cubic

0.5369

0.9799

SVR_RBF

0.6662

0.9811

FFNN

0.5405

0.9918

RNN

0.0007

0.9999

Reviewer 4 Report

Comments and Suggestions for Authors

1. In the paper title, it is "AI assisted RFID-based sensor" but there are insufficient discussion on AI techniques.

2. There are numrous reported research efforts on AI-assisted RFID-based sensor. It would be great to discuss novelty of the proposed research compared to other works.

3. Could you show efficiency of the RF energy harvesting? It would be great to measure collected DC power from the source antenna.

4. Show measurement setup for the proposed RFID-based sensor.

Author Response

  • Responses to Reviewer 4

Comments to the Author

This paper has been revised by the authors, but the quality doesn't meet the relative requirements of this transactions, so major revision should be done for this version of the paper strictly according to the following suggestions and then re-submit to this journal or the other journals:

  • Recent More achievements of this topic on An RFID-Powered Multi-Sensing Fusion Industrial IoT System for Assessment and Sensing should be added for section 1. The new idea should be highlighted in the abstract and introduction.

Comments are well taken. The abstract and introduction are revised to highlight the main novelty and contribution of this work.

  • In the paper title, it is "AI assisted RFID-based sensor" but there are insufficient discussion on AI techniques.

Comment is well taken. First, the used AI model is discussed in more detail in Section 5. Furthermore, this paper primarily discusses the advantages of applying RFID tag-integrated sensors in manufacturing lines and supply chains, provides guidelines for system/sensor design, and conducts parameter studies, among other considerations. The introduction of AI methods is to demonstrate that the multi-sensing and data traceability of RFID tag-integrated sensor and machine learning methods can provide effective QAS approach. A too in-depth discussion of AI technology is somewhat beyond the scope of this article. In the revised manuscript, the used regression models have been discussed in more detail. The revised content is shown in red text.

  • There are numrous reported research efforts on AI-assisted RFID-based sensor. It would be great to discuss novelty of the proposed research compared to other works.

The introduction and Section 5 are both revised to demonstrate the novelty of this work compared with existing investigations. Table 3 shows the comparison of this work with other RFID sensors. This work concern more about the hardware/system design of RFID sensing and the integration of machine learning with RFID sensing data. Please check the revised Section 1 and Section 5 for more details.

  • Could you show efficiency of the RF energy harvesting? It would be great to measure collected DC power from the source antenna.

Comment is well taken. The efficiency results are shown in Figure 14 (b).

  • Show measurement setup for the proposed RFID-based sensor.

Comment is well taken. The measurement setup for passive and semi-passive RFID tag-integrated sensor are shown in Figure 14 and Figure 15 (a), respectively. As shown in Figure 14, the RF energy harvester performance of the passive design is measured. Figure 15 shows the food product measurement setup.

Round 2

Reviewer 1 Report

Comments and Suggestions for Authors

sensors-2870728

Artificial Intelligence Assisted RFID Tag-integrated Multi-sensor for Quality Assessment and Sensing

Thank you for allowing me to revise the resubmitted manuscript titled " Artificial Intelligence Assisted RFID Tag-integrated Multi-sensor for Quality Assessment and Sensing." I believe the submitted manuscript and presented work is suitable for publishing in the Sensors.

Author Response

Thank you for allowing us to revise the resubmitted manuscript entitled "Artificial Intelligence Assisted RFID Tag-integrated Multi-sensor for Quality Assessment and Sensing.". The revision has been made accordingly. We believe the submitted manuscript is now suitable for publishing in the Sensors.

Reviewer 4 Report

Comments and Suggestions for Authors

Most of comments are addressed.

Please update figure resolution and size of font in Figures.

It is hard to read data in Fig.6, 7, etc.

Author Response

Figures 4, 5, 6, 7 are revised. The resolution and size of font are increased to make them clear.